# Understanding the Influence of Socioeconomic Variables on Medicinal Plant Knowledge in the Peruvian Andes

**DOI:** 10.3390/plants11202681

**Published:** 2022-10-12

**Authors:** Fernando Corroto, Oscar Andrés Gamarra Torres, Manuel J. Macía

**Affiliations:** 1Instituto de Investigación para el Desarrollo Sustentable de Ceja de Selva, Universidad Nacional Toribio Rodríguez de Mendoza de Amazonas, Chachapoyas 01001, Peru; 2Departamento de Biología, Área de Botánica, Universidad Autónoma de Madrid, Calle Darwin 2, ES–28049 Madrid, Spain; 3Centro de Investigación en Biodiversidad y Cambio Global (CIBC-UAM), Universidad Autónoma de Madrid, Calle Darwin 2, ES–28049 Madrid, Spain

**Keywords:** biocultural conservation, ecosystem services, livelihood, local botanical knowledge, medicinal plants, quantitative ethnobotany, sustainability

## Abstract

In this study, we analyze the impact of 18 socioeconomic factors at individual, family, and locality levels to understand their influence on medicinal plant knowledge (MPK) in four provinces and 12 localities of the northern Andes of Peru. We interviewed 50 participants per locality (totaling 600 people) from lowlands and highlands ecoregions. The participants were balanced in terms of generations and gender. We performed multivariate statistical analyses—generalized linear mixed models (GLMMs) and nonmetric multidimensional scaling (NMDS) ordinations—that showed the influence of each socioeconomic variable on the medicinal plant knowledge of people in the different sites. At the individual level, we found that most participants with higher MPK were women, elders, people with lower levels of education and job qualifications, non-migrants, and participants who have lived for a long period in the same region. At the family level, we found that participants living in low economic conditions with few material goods, including their means of transport, tools possession, access to technology, farm size, number of farm animals, and house quality, had higher MPK. At the locality level, we found that people living in more isolated areas with scarce regional services, such as access to paved roads, hospitals, big markets, tourist development, and chlorination of drinking water, had higher MPK. In short, people with less access to modern services and with low economic resources are the main depositaries of MPK. Policy makers and decision makers should consider the significance of MPK in alleviating health problems and diseases in Andean regions, especially for people with rural livelihoods. This local botanical knowledge of medicinal plants should be preserved in the area as a great natural heritage for humanity.

## 1. Introduction

Socioeconomic and cultural changes have different implications for the medicinal plant knowledge (MPK) of distinct human populations, at both the global and local scales. MPK is particularly significant in developing countries that conserve traditions more prominently and depend on natural resources [1,2]. In recent decades, the socioeconomic framework has changed rapidly during the lifetimes of people in developing countries [3]. Usually, when the socioeconomic situation has improved at both personal and regional levels, it has been accompanied by the loss of MPK and cultural identity for local people [4,5].

One way to analyze these socioeconomic and cultural changes is through ethnobotanical studies, which provide good models for understanding patterns of change across societies, focusing on the use of plants by people who clearly depend on their socioeconomic and cultural situations [6,7,8]. In this sense, medicinal plants have a special relevance for economies and for the conservation of culture, as they help in maintaining health, providing basic sustenance for people’s livelihoods, and reflecting their traditions, at costs that are cheaper than those of allopathic medicine [9,10].

Socioeconomic variables can be classified into three different levels for a human group living in a rural environment: an individual level, a family level, and a locality level [11]. All three of these three levels have been reported to influence MPK. 

At the individual level, many earlier studies documented that women and older age groups are the most important custodians of MPK [12,13]. In addition, there are other variables that influence MPK at this level, such as higher levels of education, which seem to displace the traditional culture, knowledge, and skills of younger people [14], and personal occupations, as people in highly specialized jobs demonstrate a low level of MPK [15,16]. The migratory status and time-of-residence of people in a particular area are also related to MPK, which is more common among residents who were born and lived in the same region for a long time [17,18].

At the family level, distinct variables have also been used to analyze different forms of economic wealth in relation to MPK, such as means of transport, access to technological services, and ownership of tools, cattle, and farmland. In all these situations, having more possessions is related to lower MPK [19]. If a family home is in good condition, this can be related to the family’s greater economic ability to repair any damage to the home; such families tend to have lower MPK [20].

At the locality level, low access to regional services, such as paved roads, hospitals, and big markets, results in the isolation of a locality, enhancing the preservation of their MPK [21,22]. In contrast, the implementation of water treatment with chlorination for human consumption reduces the incidence of diarrheal diseases and other infectious diseases transmitted by water [23], which may be related to a low transmission of MPK from the elder people to younger people. Likewise, tourist attractions provide direct economic benefits for local inhabitants, which may be an advantage for economic development but a disadvantage for maintaining traditional knowledge [24].

In Peru, some studies have focused on the maintenance of MPK at the personal level [25,26,27,28], but to the best of our knowledge, this is the first study that analyzes the relationships between the socioeconomic situation of people at individual, family, and locality levels and their usage of medicinal plants. The aim of this paper is to combine a study that integrates the largest number of socioeconomic factors, gathered from hundreds of participants living in different Andean ecoregions, with their MPK obtained over the course of their lives, using a stratified interview methodology with the collaboration of local mestizo inhabitants.

Specifically, in this study we analyzed the impact of socioeconomic factors on the use of medicinal plants for rural populations in the Peruvian Andes. Our first hypothesis, related to the influence of socioeconomic factors at the individual level, was that people with higher educational levels, specialized jobs, shorter residence times, and migrant status will have lower MPK. Our second hypothesis, related to the impact of socioeconomic variables at the family level, was that people with greater financial resources, in the form of modern means of transport and tools, access to technological services, possession of livestock, and farm size, will know less about medicinal plants and their uses. Our third hypothesis, at the locality level, was that people living in localities that are close to tourist attractions, paved roads, hospitals, big markets, and treated water services will have lower MPK.

## 2. Results

### 2.1. The Influence of Socioeconomic Factors on MPK

Overall, localities in the highlands had higher numbers of medicinal plants and reported greater usage of medicinal plants, compared with localities in the lowlands, particularly in the province of Chachapoyas (Table 1). In relation to the age of participants and family size, the results showed a positive relationship with MPK in all highlands and lowlands, although the significance of this relationship differed between them.

At the individual level, the results for the different socioeconomic factors were as follows (Figure 1, Appendix A): (1) Women had MPK in higher numbers than men in the four provinces, although in the locality of Longuita the results were reversed (Figure 1A); (2) Overall, education had a negative effect on MPK, indicating that MPK decreased as formal education increased in all localities, with the exceptions of Valera and Totora (Figure 1B); (3) People with specialized jobs demonstrated lower MPK than people with basic occupations, with the exception of Cuispes (Figure 1C); (4) Non-migrants had MPK in higher numbers than migrants, with the exceptions of Huambo and Longuita (Figure 1D); and (5) Participants who had been living in an area for longer periods had higher MPK, with the exceptions of Huambo and Longuita (Figure 1E).

At the family level, we found the following results (Figure 2, Appendix A): (1) Participants that possessed no transport or basic transports demonstrated higher MPK than participants with motor vehicles (Figure 2A); (2) Participants who did not possess any tools, or just basic tools, had higher MPK than participants with semi-automatic tools, with the exception of Longuita (Figure 2B); (3) People in localities of the lowlands, with lower MPK, had access to cable TV or the Internet, with the exception of Cuispes, while people in localities of the highlands, with higher MPK, had no access to technology, with the exception of Longuita (Figure 2C); (4) Participants who did not possess farm animals. or possessed a low number of them, had higher MPK than participants who used farm animals for trade, with the exceptions of Santa Rosa and Longuita (Figure 2D); (5) Families with a farm size of less than five hectares had higher MPK, with the exception of San Carlos and Huambo in the lowlands and Quinjalca and Longuita in the highlands (Figure 2E); (6) Families whose houses had major defects had higher MPK than those with good quality houses, with the exception of Cuispes, Huambo, and Santa Rosa in the lowlands (Figure 2F).

At the locality level, we found that participants living in localities without access to paved roads, hospitals, big markets, and tourist attractions showed higher MPK than participants living in localities with access to these services (Table 2). In addition, people from localities that had no chlorination system for drinking water had higher MPK than people from localities that had this service.

### 2.2. Mixed-Model Effects of Socioeconomic Factors on MPK

We found that nine of the 18 socioeconomic factors showed a significant association with MPK in 11 of the 12 localities, with the exception of María (highlands), and those 11 localities showed significant associations with one to three socioeconomic factors (Table 3). At the individual level, the associations were as follows: (1) Gender had a statistically significant negative association in three localities of the lowlands (San Carlos, Valera, and Totora) and in one locality of the highlands (Quinjalca); (2) Education had a statistically significant negative influence on MPK in three localities of the lowlands (San Carlos, Santa Rosa, and Totora) and in one locality of the highlands (Yomblón); (3) Occupation had a negative association with MPK in two localities of the lowlands (San Carlos and Santa Rosa); (4) Migration had a statistically significant negative relationship with MPK only in Longuita; (5) Time-in-residence showed significant associations with MPK in two localities of the highlands (Longuita and Yomblón). 

At the family level, the associations were as follows: (1) Family size had a statistically significant positive association in three localities of the highlands (Granada, Olleros, and Yomblón); (2) Transport showed statistically negative associations with the maintenance of MPK only in Totora; (3) The possession of tools had a negative association with MPK in two localities of the highlands (Granada and Olleros) and two localities of the lowlands (Cuispes and Valera); and (4) Access to technological services showed negative significant associations in the lowlands (Huambo and Santa Rosa) and in the highlands (Granada).

### 2.3. Relationship between Socioeconomic Factors and Medicinal Categories

Overall, localities in both the highlands and the lowlands tended to be grouped into different medicinal categories. In the highlands, the three localities of the Chachapoyas province showed a clear group in each of the different medicinal categories. Within the localities of the Luya province (highlands), Yomblón tended to be closer to the localities of the Chachapoyas province, while Longuita and María were grouped across all the medicinal categories. In the lowlands, the three localities of the Rodríguez de Mendoza province were grouped for most of the categories (Figure 3A,C,E–G), but Huambo was clearly separated in the categories of s”Skin and subcutaneous tissue” and ”Nervous system and mental health” (Figure 3D,H, respectively). Finally, the localities of the Bongará province were grouped for most of the medicinal categories (Figure 3A–E,H), but Valera showed a distinct pattern for the categories of “General ailments and unspecific symptoms” and ”Pregnancy, birth and puerperium” (Figure 3F,G, respectively).

## 3. Discussion

### 3.1. Influence of Socioeconomic Factors on MPK at the Individual Level

The results for the socioeconomic factors at the personal level supported our first hypothesis: higher educational levels, higher job qualifications, shorter residence times in a region, and non-migrant status were negatively related to the maintenance of MPK. These findings can be explained on the basis that conventional education currently removes young people from their natural environment and, consequently, reduces their opportunity to learn traditional knowledge from their predecessors [29,30]. Education has a close relationship with future occupation, because achieving a higher level of education is related to having a specialized job; such jobs are usually found in cities with a certain concentration of businesses and/or public institutions and, therefore, city people tend to have lower traditional knowledge [31,32]. Migratory status showed a negative relationship with MPK, consistent with the commonly reported phenomenon that migration and globalization are related to the loss of MPK [33]. Time-in-residence showed a pattern similar to that of migratory status, indicating that longer residence times for people in a region generate more direct contact with their environment and, therefore, time-in-residence is related to higher MPK [1,34], although some other studies have found an opposite pattern [18,35].

Generally, women had higher MPK than men, consistent with most earlier studies in the Andes, based on their role in taking care of children and elders [13,36]. Overall, elders showed greatest MPK, as they had the opportunity to accumulate traditional knowledge in earlier times, as many other papers have previously documented [28,37].

### 3.2. Influence of Socioeconomic Factors on MPK at the Family Level

Our second hypothesis was also supported. We established that the possession of material goods (modern means of transport and tools, access to technological services, possession of livestock, and farm size) is related to lower MPK, and our results support that conclusion in most localities. The possession of assets depends on the availability of economic resources to obtain them, and their use in close relation to agricultural and livestock practices may be focused on trade; consequently, such usage may require more technical tools, better means of transport, and access to technological services [38,39]. This hypothesis was also related to the number of children per family who had a positive association with the maintenance of MPK in both ecoregions. Families with a larger number of members tend to have lower economic possibilities and more economic difficulties in accessing health services, as previously reported in other studies [40,41]. In general, families living in houses that needed some repair showed higher MPK than families living in houses of good quality, which again reflects their respective economic resources (Corroto, personal observations).

### 3.3. Influence of Socioeconomic Factors on MPK at the Locality Level

Our results support our third hypothesis, confirming that participants living in localities close to tourist attractions, paved roads, hospitals, big markets, and water chlorination showed a lower MPK. Highlands localities had fewer regional services than lowlands localities, and this isolation provided an advantage for the conservation of medicinal traditional knowledge [22,42,43]. In contrast, the lowlands localities are characterized by better levels of regional economic development, such as paved roads, commercial urban centers, and health services, which are related to a lower maintenance of MPK [44,45]. In addition, localities close to hospitals tend to place their trust in allopathic medicine, unlike those distant localities that still trust in traditional medical healers, as is the case with people from the highlands in our study [46,47]. The localities closer to tourist attractions have medical centers and posts, which are related to the progressive abandonment of traditional medicine [48]. Finally, all localities in the lowlands, unlike some localities in the highlands, have a chlorinated water supply for human consumption, which is an indicator of progress in developing countries. as its implementation reduces health problems [49]; therefore, the people in the localities of Chachapoyas, which have no water chlorination systems, showed higher MPK.

### 3.4. Similarities of Socioeconomic Factors across Localities and Medicinal Categories

Based on socioeconomic factors, the different ecoregions and localities showed a relatively similar pattern of spatial ordination for the different medicinal categories. The separation into two groups of the highlands localities was related to their differential socioeconomic factors. For most medicinal categories, the three localities of the Chachapoyas province were grouped together with the locality of Yomblón (in the Luya province), because they share geographical isolation and limited access to regional services, including paved roads. Their isolation and limited access to contemporary health care favor the preservation of traditional medical practices that are similar, as reflected in the different medicinal categories [50]. In the case of lowlands, the localities were mainly grouped within the same province, with two exceptions in terms of divergent medicinal categories: Valera (in the Bongará province), which is closer to the localities of the Rodríguez de Mendoza province. and Huambo (in the Rodríguez de Mendoza province), which is closer to the localities of the Chachapoyas province. These localities were grouped together because they have analogous socioeconomic factors and geographical situations within their regional context. Finally, one of the factors that most affected the maintenance of MPK was probably proximity to a hospital, which may explain that the localities of the Rodríguez de Mendoza province suffered a greater loss of MPK, as has been recorded in other studies [51].

## 4. Materials and Methods

### 4.1. Study Area

Our research was carried out in high tropical montane regions (highlands) and low tropical montane regions (lowlands) of the northeastern Peruvian Andes, in the Amazonas Department (Figure 4). These classifications were based on the descriptions of the Peruvian ecological regions proposed by León et al. [52]. In each ecoregion, we studied two provinces (Chachapoyas and Luya in the highlands, Bongará and Rodríguez de Mendoza in the lowlands) and three localities per province, all of which were inhabited by mestizo people. Six of the 12 localities were established between 2500 m and 3500 m in the highlands (Granada, Olleros, and Quinjalca in Chachapoyas; Longuita, María, and Yomblón in Luya), while the six other localities were located between 1500 m and 2500 m in the lowlands (Cuispes, San Carlos, and Valera in Bongará; Huambo, Santa Rosa, and Totora in Rodríguez de Mendoza). In both ecoregions, there is a seasonal climate that alternates with a wet season between November and May, and a dry season for the remaining months of the year, with an average annual rainfall of 780 mm and average temperatures of 14 °C, which are somewhat lower in the highlands, compared with the lowlands (900 mm and 19 °C, respectively) [53].

The selection of the localities was based on different geographical characteristics and socioeconomic development. In general terms, the populations of the lowlands have more economic resources and better infrastructures than the populations of the highlands (Table 4). The localities of the Rodríguez de Mendoza province (lowlands) are the only ones that have a productive coffee monoculture and intensive livestock, while the three other provinces have subsistence agriculture (Bongará) or more diverse productive agriculture (highlands). There are touristic attractions in both Bongará (lowlands) and Luya (highlands), with the exception of Yomblón. Localities in the Rodríguez de Mendoza and Luya provinces have paved roads (except for Yomblón). The only localities with a hospital are in Rodríguez de Mendoza. The localities of the Chachapoyas province are the only ones with no access to big markets or water chlorination. The lowlands localities have a higher population density than the highlands localities; Chachapoyas, by far, is the province with the lowest population [30].

### 4.2. Data Collection

A total of 600 semi-structured interviews (50 per locality) were carried out to gather information on medicinal plants. We divided participants into five age groups (18–30, 31–40, 41–50, 51–60, and >60 years old), balancing the participants between men and women (see [55] for further details). We conducted closed interviews with all of the participants to gather socioeconomic information related to 18 variables, classified into three levels: an individual level, a family level, and a locality level (Table 5). At the individual level, we included gender, age, education, occupation, migratory status, and time-of-residence in the region. At the family level, we studied family size, means of transport, possession of tools, access to technological services, number of farm animals, farm size, and house quality. At the locality level, we analyzed the proximity to paved roads, hospitals, big markets, tourist attractions, and water treatment for human consumption.

The identification of plant specimens was carried out in the Herbarium Truxillense (HUT), and a voucher containing all of the collected information was deposited in this institution. The scientific names followed the plant list [56] and the taxonomic classification at the family level followed the angiosperm phylogeny group [57].

### 4.3. Data Analysis

All of the reported medicinal indications were classified into 19 use categories, following international standards [58], with added modifications based on the cultural, ritual, or magical diseases proposed by [59,60].

We classified the 18 socioeconomic factors into three types of variables: nominal (gender and locality); ordinal (education, occupation, migration status, time-in-residence, transport, tools, access to technological services, farm animals, farm size, house quality, proximity to paved roads, proximity to hospitals, proximity to big markets, proximity to places of tourist interest, and water quality); and continuous (age and family size) (Table 5).

Two ethnobotanical indicators were analyzed: (1) reports on the usage of medicinal plants use, which were the sum of all medicinal uses reported by one participant for all of the species known, and (2) useful medicinal plant species, which corresponded to the sum of all of the documented species by each participant. We defined “medicinal use” as the use of a plant part of a species that is associated with a medicinal category for a particular disease or ailment [40]. We found strong correlations between the two ethnobotanical indicators for all localities (highlands r = 0.66–0.84; lowlands r = 0.62–0.82) and, therefore, we decided to use only the reports on medicinal plants usage as the dependent variable in all of the the subsequent analyses.

To analyze MPK related to the 18 socioeconomic variables studied in the 12 localities, we first used multivariate analysis of variance (MANOVA) analyses with the full dataset and its corresponding post hoc Tukey test for the 16 categorical variables and Pearson correlations for the two continuous variables (Table 5). After these analyses, we selected the socioeconomic variables for use in the analyses that followed. We excluded categorical variables that had significant differences based on their different levels, and also excluded continuous variables with *p* < 0.05.

To assess the variations in traditional knowledge in the 12 studied localities of the Andean lowlands and highlands, we carried out generalized linear mixed models with negative binominal distribution to analyze the effects of the socioeconomic variables on MPK, using the reports on medicinal plants usage. We implemented models for each of the three different socioeconomic levels: an individual level, a family level, and a locality level. The dependent variable was the number of reports on medicinal plants usage, and the independent variables with a fixed effect were the socioeconomic factors that were selected in each region, based on the initial descriptive analysis. Regions were established as random factors for the categorical variable locality. The one-level random-intercept model that we constructed was based on the following formula:
$$Y_{ij}=ƴ00+(\beta X_{i}\ldots .)+(\tau X’i\ldots .)+r0j+\sigma L+eij$$
where *Yij* is the independent variable, ƴ00 is the common intercept, *β* and τ are the respective coefficients of the continuous variables *Xi* and categorical *X´i*, *r0j* has a normal distribution with median 0, standard deviation *σL* represents the variability of the 12 localities, and *eij* is the error or residual for each of the interviewees.

To visualize whether the provinces and localities were spatially grouped according to socioeconomic levels, we performed nonmetric multidimensional scaling (NMDS) analyses with all socioeconomic variables for the eight most-cited medicinal categories (>800 reports of usage). To calculate the (dis-)similarities between the localities, we used Euclidean distances and we implemented analyses for the first two axes that were subsequently plotted in Figure 3A–H. All of the analyses were performed with Infostat Analytical Software for Windows [61].

## 5. Conclusions

Socioeconomic factors have a strong influence in the maintenance of medicinal plant knowledge (MPK) along the northern Peruvian Andes. Our research showed that people with higher MPK have lower levels of education and unspecialized jobs, and that they have been living in the same region for a long period of time. Such people usually have large families, low economic wealth, and few material goods, and they live in localities that are relatively isolated, with scarce regional services. In short, people with less access to modern services and with low economic resources are the main carriers of MPK. This conclusion can probably be generalized to other developing countries, in similar Andean regions, for all types of traditional knowledge. Politicians and decision makers who are concerned with social and conservation issues, at both national and international levels, should consider the significance of MPK when implementing poverty alleviation programs, in order to improve people’s livelihoods. In addition, this magnificent traditional knowledge must be documented for humankind before it vanishes forever.

## Figures and Tables

**Figure 1 plants-11-02681-f001:**
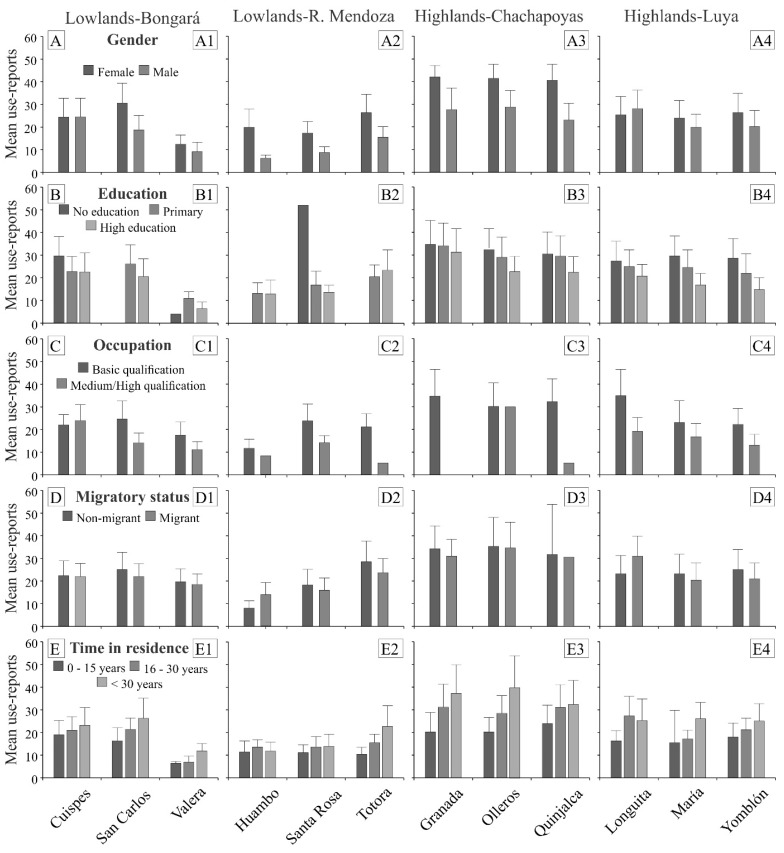
Relationship between socioeconomic factors at the individual level and MPK, measured as the mean of reports of medicinal usage gathered in 12 localities of the northern Peruvian Andes.

**Figure 2 plants-11-02681-f002:**
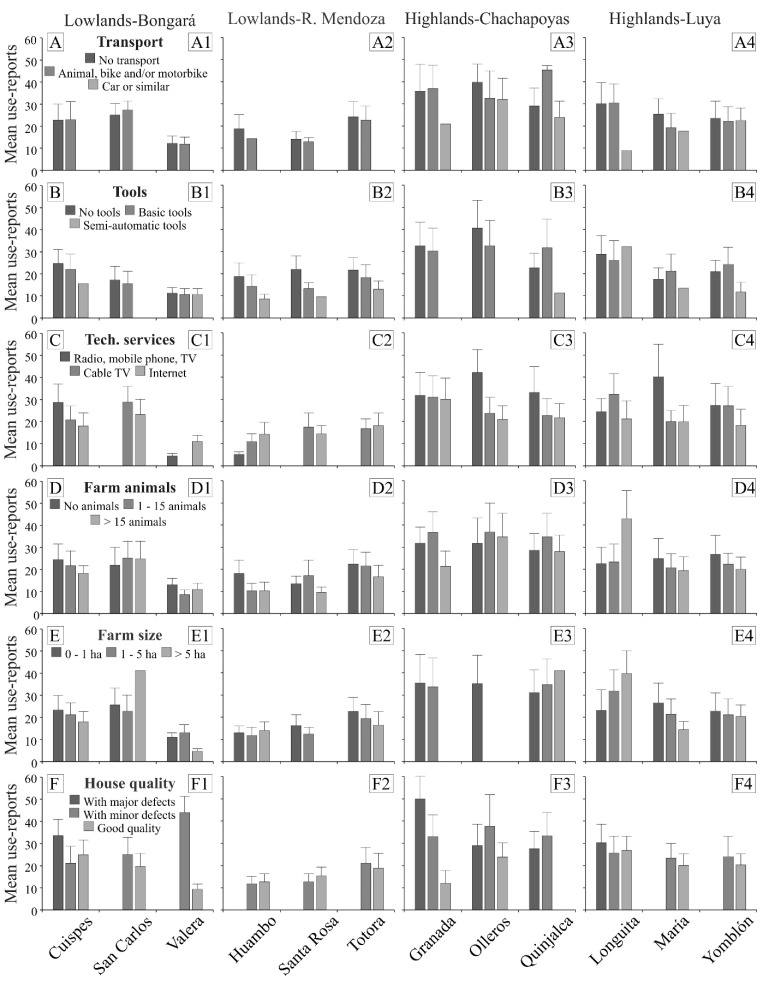
Relationship between socioeconomic factors at the family level and MPK, measured as the mean of reports of medicinal usage gathered from participants in 12 localities of the northern Peruvian Andes.

**Figure 3 plants-11-02681-f003:**
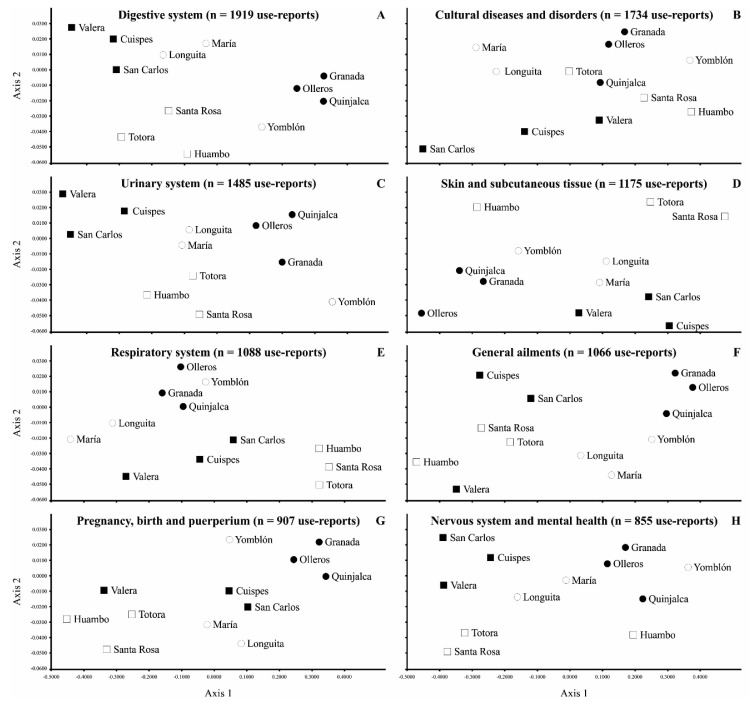
Nonmetric multidimensional scaling (NMDS) ordination showing the similarity of the 18 socioeconomic factors influencing MPK (means of reports of medicinal usage) for the eight most cited medicinal categories in 12 localities of the northern Peruvian Andes. Symbols indicate the different provinces, as follow: Lowlands: Bongará (■) and Rodríguez de Mendoza (□). Highlands: Chachapoyas (●) and Luya (○).

**Figure 4 plants-11-02681-f004:**
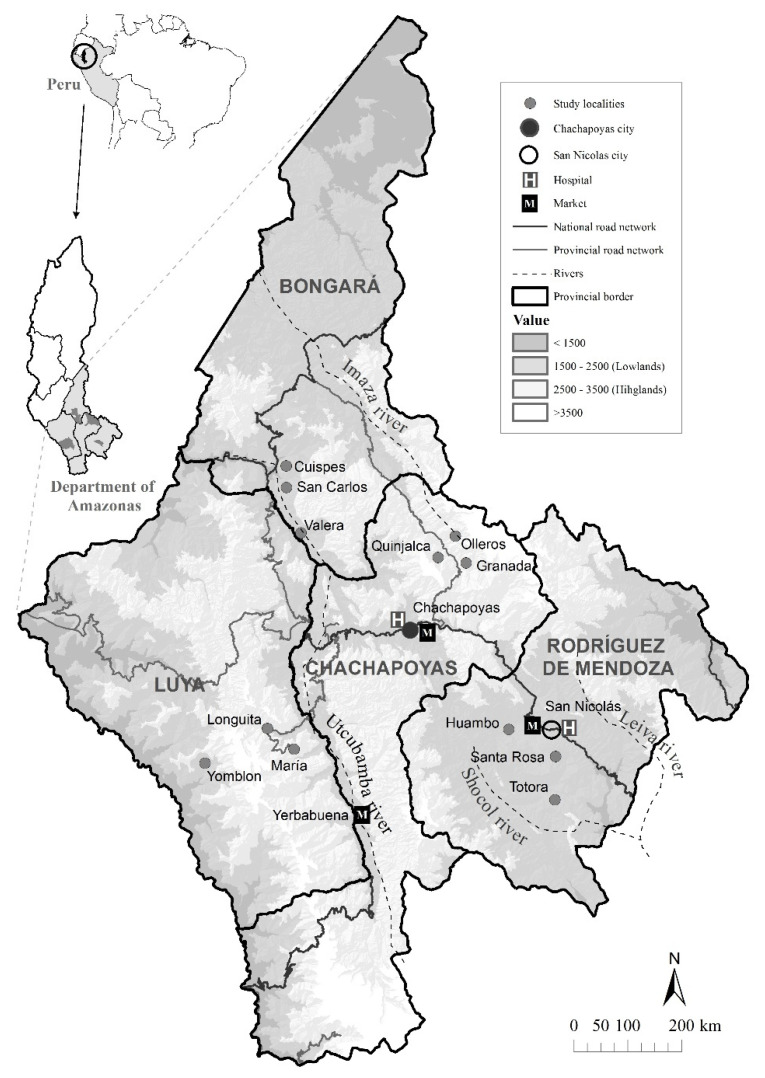
Map of the study area in the northern Peruvian Andes, showing the two ecoregions (highlands and lowlands), the four provinces, and the 12 localities where information about medicinal plants uses and socioeconomic factors was obtained via 600 interviews.

**Table 1 plants-11-02681-t001:** Mean (± standard deviation) values of the number of medicinal plant species and the reported usage of medicinal plants, based on information gathered from participants and mean Pearson correlation statistics for two socioeconomic variables and MPK (based on the reported uses). We carried out 600 interviews in 12 localities of the Peruvian Andes.

Ecoregion—Province	Locality	# Medicinal Species	# Medicinal Plant Usage	Age	Family Size
Lowlands—Bongará	Cuispes	19.4 ± 6.4	22.2 ± 7.1	0.08	0.14
	San Carlos	22.0 ± 5.8	24.6 ± 7.9	0.17	0.03
	Valera	9.6 ± 3.6	10.8 ± 4.5	0.14	0.28
Lowlands—R. Mendoza	Huambo	10.6 ± 3.9	12.3 ± 4.1	0.24	0.04
	Santa Rosa	13.1 ± 4.0	13.8 ± 4.2	0.15	0.19
	Totora	18.5 ± 4.4	20.8 ± 5.2	0.09	0.14
Highlands—Chachapoyas	Granada	26.9 ± 7.8	34.8 ± 8.9	0.23	0.28
	Olleros	27.2 ± 7.3	35.1 ± 7.8	0.29	0.09
	Quinjalca	26.8 ± 7.5	31.8 ± 7.6	0.06	0.11
Highlands—Luya	Longuita	22.4 ± 6.6	26.2 ± 7.1	0.12	0.13
	María	19.0 ± 5.9	22.3 ± 6.6	0.12	0.25
	Yomblón	20.1 ± 6.0	23.1 ± 6.8	0.19	0.18

**Table 2 plants-11-02681-t002:** Relationship between socioeconomic factors at the locality level and MPK, measured as mean (± standard deviation) of reports of medicinal usage gathered from participants in 12 localities of the northern Peruvian Andes. The impact of the distinct socioeconomic factors per locality was classified at three levels (high, medium, and low), depending on the proximity to the analyzed factor and the availability of chlorinated water.

Attributes	High	Medium	Low
Proximity of tourist attractions	19.7 ± 6.4	23.4 ± 7.2	30.1 ± 8.4
Proximity to paved roads	19.1 ± 6.7	19.2 ± 6.4	31.2 ± 8.2
Proximity to hospitals	15.6 ± 5.4	21.2 ± 6.8	31.2 ± 8.2
Proximity to big markets	17.4 ± 5.6	24.2 ± 7.5	31.2 ± 8.2
Chlorination of drinking water	18.9 ± 6.1	–	29.1 ± 8.0

**Table 3 plants-11-02681-t003:** Mixed-model effects of the number of reports of medicinal use and coefficients of the socioeconomic factors evaluated in 12 localities of northern Peruvian Andes. Levels of significance: * *p* < 0.05; ** *p* < 0.01; nd: no data.

Attributes	Lowlands—Bongará			Lowlands—Mendoza			Highlands—Chachapoyas			Highlands—Luya		
	Cuispes	San Carlos	Valera	Huambo	Santa Rosa	Totora	Granada	Olleros	Quinjalca	Longuita	María	Yomblón
Intercept	45.462	24.553	35.234 **	11.791	23.797	34.316 *	37.979	24.263	11.641	19.558	42.486	23.159
Women	1.245	3.244 **	2.506 *	0.821	1.109	2.068 *	1.110	1.080	2.471 *	0.809	0.908	0.859
Education	1.077	−3.618 *	0.714	−1.752	−4.291 **	−2.662 *	−0.580	0.490	0.520	−1.532	−0.400	−3.246 *
Occupation	1.012	−4.783 **	−1.527	0.206	−4.935 **	−1.672	nd	0.919	−0.353	−1.992	−0.506	0.065
Migratory status	−0.373	−0.348	−0.161	0.508	0.422	0.534	−0.161	−1.029	−0.015	−3.347 *	−1.656	−0.107
Time-in-residence	0.927	0.791	0.734	0.839	0.248	1.449	0.811	1.599	−0.143	2.471 *	1.711	2.124 *
Family size	0.697	0.409	0.218	0.703	1.405	0.616	2.072 *	2.782 *	0.979	1.405	0.779	3.271 *
Transport	0.863	0.919	−0.969	−0.389	−0.835	−2.138 *	0.753	−0.536	0.767	−1.596	−0.901	−1.528
Tools	−3.927 *	−1.207	−4.768 *	−0.361	−0.738	−0.424	−4.284 *	−8.122 **	0.326	0.951	1.298	−0.966
Technologic access	−0.406	−0.073	−1.589	−1.771 *	−2.741 **	0.053	−2.195 *	−0.252	−1.055	0.857	−0.238	−0.548
Locality	8.682	5.466	6.022	4.440	8.212	6.799	6.245	3.599	2.421	3.328	7.056	5.592
Interviewed residuals	6.324	3.026	4.073	1.893	5.889	3.802	3.935	1.447	0.971	1.722	4.800	3.232

**Table 4 plants-11-02681-t004:** Socioeconomic characteristics of the two studied ecoregions in the tropical montane areas of the northern Peruvian Andes. The population data were obtained from the Instituto Nacional de Estadística e Informática (INEI) [54] and referred to the number of inhabitants per province.

Attributes	Bongará (Lowlands)	R. Mendoza (Lowlands)	Chachapoyas (Highlands)	Luya (Highlands)
Agriculture	Subsistence agriculture (corn, fruit trees, and pasture)	Predominance of productive coffee	Predominance of productive Andean tubers, corn, and pasture	Predominance of productive Andean tubers, corn, and pasture
Livestock	Extensive subsistence cattle	Intensive bovine cattle and swine	Extensive subsistence cattle	Extensive subsistence cattle
Fisheries	Small-scale trout farms	Fishing in rivers	Small-scale trout farms	Small-scale trout farms
Tourist attractions	Yes	No	No	Yes
Paved roads	No	Yes	No	Yes
Hospitals	No	Yes	No	No
Big markets	Yes	Yes	No	Yes
Water chlorination	Yes	Yes	No	Yes
Population	2588	3277	1034	1991

**Table 5 plants-11-02681-t005:** Description of the 18 socioeconomic variables for which data were obtained from 600 interviews in 12 localities of the northern Peruvian Andes.

Independent Variable Name	Variable Scale	Variable Type	Variable Classification
Gender	Individual	Nominal	(0) Men; (1) Women
Age (years)	Individual	Continuous	Between 18 and 91
Education	Individual	Ordinal	(0) No education; (1) Primary education; (2) High education
Occupation	Individual	Ordinal	(0) Basic qualification; (1) Medium/High qualifiation
Migratory status	Individual	Ordinal	(0) Non-migrant; (1) Migrant
Time-in-residence	Individual	Ordinal	(0) Between 0–15 years; (1) Between 16–30 years; (2) More than 30 years
Family size	Familiar	Continuous	Between 0–11 children
Means of transport	Familiar	Ordinal	(0) No transport; (1) Animal and bicycle; (2) Motorbike, car, or similar
Tools	Familiar	Ordinal	(0) No tools; (1) Basic tools; (2) Semi-automatic tools
Technological services	Familiar	Ordinal	(0) Radio, mobile phone, TV; (1) Cable TV; (2) Internet
Farm animals	Familiar	Ordinal	(0) No animals; (1) 1–15 animals; (2) >15 animals
Farm size	Familiar	Ordinal	(0) 0–1 ha; (1) 1–5 ha; (2) >5 ha
House quality	Familiar	Ordinal	(0) Some defects; (1) Good quality, no defects
Proximity to tourist attractions	Locality	Ordinal	(0) Less than 1 h; (1) 1–2 h; (2) More than 2 h
Proximity to paved roads	Locality	Ordinal	(0) Less than 1 h; (1) 1–2 h; (2) More than 2 h
Proximity to hospitals	Locality	Ordinal	(0) Less than 1 h; (1) 1–2 h; (2) More than 2 h
Proximity to big markets	Locality	Ordinal	(0) Less than 1 h; (1) 1–2 h; (2) More than 2 h
Water quality	Locality	Ordinal	(0) Drinking water with a chlorination system; (1) Unguarded, without a chlorination system

## Data Availability

The data presented in this study are available in this article and its Appendix A.

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
