# Peer review of "Understanding the Influence of Socioeconomic Variables on Medicinal Plant Knowledge in the Peruvian Andes"

_plants, 2022, doi:10.3390/plants11202681_

Round 1
Reviewer 1 Report
Please find attached my comments.
More important introduction and methods demand further work.
-the introduction needs bettern explanation of the study motivation and the reasons that lead to the hypothesis.
-the methods are poorly described. The design is good and the data is impressive. However, the modelling lacks a lot of the details.
-the methods indicate that a GLM was done, but the results provide the output of a GLMM. Both models have completely different structures. Seen that you used mixed models (Table 3). So what were the fixed and random terms?
-Moreover, it is unclear how many models were done, if different models were done per level, per localty or per province (or for all combinations)
-how to acess model significance? Pseudo R2, comparission with Null models using AICs? Somehow this should be presented so the reader can see if your results are significant despite of the outcomes in model summary (variables significative or not). Without this assessesment it is not possible to tell if H1 holds.
-Did you use negative binomial distributions for every model? What if a model happen to have only continuous variables (e.g. age or family size)? Then this distribution would not be the most adequated one.
-Colinearity is briefly described and needs bettern explaination. Why did you not used VIF but prefered to test correlations a priori? How did you chose the variables that were excluded from the modelling?
-No description of the NMDS analysis is given. What was the distanced used to calculate the (dis)similarities? Bray-curtis? Euclidian? What was the stress or the percentage explained by each axis? Did you run the NMDS for only 2 axis or for more axis and the picked the first two?

Author Response
Responses to Reviewer 1 are in the attached Word file

Reviewer 2 Report
Dear authors, I have read Your article with interest. The research is a big survey. However, I also found the issues which require explanations:
For me the presentation of the data in Table 1 is unclear:
- why Age category is presented in the interval from 0.08 to 0.29 ? what is the unit (years, months or the other value ?)
- the same comment about category - Family size:
why its shown as the interval from 0.03 to 0.28 ? it is unclear for me...
Author Response
Please see the attachment (Responses to Reviewer 2 are in the attached Word file)

Reviewer 3 Report
Dear Authors,
The Manuscript ID: plants-1846942, Titled “Understanding the role of socioeconomic variables on medicinal plant knowledge in the Peruvian Andes” is well-structured. It states the purpose of the research, the principal results and conclusions.
Based on the evaluation of its originality, significance of content, scientific soundness, and interest to readers, a minor revision is suggested before the article may be considered for acceptance. Specific suggestions and comments are provided below.
The title is informative. It states briefly the object of the research.
The Abstract is factual. The main results of the study are included and a conclusion is defined. However, the aim of the research should be clearly declared. In addition, the methods that are used are not announced and they should be added too.
The Introduction presents data concerning socioeconomic influence on the medicinal plants knowledge (MPK) at different degrees. Three hypotheses are supposed.
The literature survey is wide-ranging. Nevertheless, I would suggest and recommend to include some missing sources:
· Toda M, Salgado E. L. R, Masuda M. Assessing medicinal plants as the linkage between healthcare, livelihood and biodiversity: a case study from native villages surrounding a second-tier city in the central Peruvian Amazon. Tropics, 2016; 25 (2): 53-65.  DOI:10.3759/tropics.MS15-07
· Mathez-Stiefel S-L, Brandt R, Lachmuth S, Rist St. Are the Young Less Knowledgeable? Local Knowledge of Natural Remedies and Its Transformations in the Andean Highlands. Human Ecology. 2012; 40(6):909-930. DOI:10.1007/s10745-012-9520-5
· Bussmann RW, Sharon D, Lopez A. Blending Traditional and Western Medicine: Medicinal plant use among patients at Clinica Anticona in El Porvenir, Peru. Ethnobotany Research & Applications. 2007; 5:185-199.
- Leatherman T L, Carey J W, Thomas R B. Socioeconomic change and patterns of growth in the Andes. 1995;97(3):307-21. doi: 10.1002/ajpa.1330970305.
These articles should be contained in the discussion too.
Material and methods are described in details, divided with subheadings according to the procedure. There is an adequacy of the methodology.
Results and discussion. The results are complete and clear. The discussion explores the significance of the results of the work based on the socioeconomic variables classified in three different levels: individual, family and locality. The work is well-illustrated. The declared hypothesizes are defensed and are consistent with the commonly reported statements and phenomenon.
The Conclusions are based on the main results..
Author Response
Please see the attachment (responses to Reviewer 3 are in the attached Word file).

Round 2
Reviewer 1 Report
The authors did a great job improving the description of the methods. I still have some minor concerns that I could detect in the text.
